# Assessment of SARS-CoV-2 Infection According to Previous Metabolic Status and Its Association with Mortality and Post-Acute COVID-19

**DOI:** 10.3390/nu14142925

**Published:** 2022-07-17

**Authors:** Alejandro de Arriba Fernández, José Luis Alonso Bilbao, Alberto Espiñeira Francés, Antonio Cabeza Mora, Ángela Gutiérrez Pérez, Miguel Ángel Díaz Barreiros, Lluís Serra Majem

**Affiliations:** 1Research Institute of Biomedical and Health Sciences, University of Las Palmas de Gran Canaria, 35001 Las Palmas de Gran Canaria, Spain; lluis.serra@ulpgc.es; 2Gerencia de Atención Primaria de Gran Canaria, 35006 Las Palmas de Gran Canaria, Spain; jalobil@gobiernodecanarias.org (J.L.A.B.); aespfra@gobiernodecanarias.org (A.E.F.); acabmorc@gobiernodecanarias.org (A.C.M.); agutperd@gobiernodecanarias.org (Á.G.P.); mdiabarb@gobiernodecanarias.org (M.Á.D.B.)

**Keywords:** total cholesterol, high-density lipoprotein cholesterol, fasting glucose levels, blood lipids, post-acute COVID-19, SARS-CoV-2, risk factors, coronavirus disease 2019, mortality

## Abstract

Background. SARS-CoV-2 infection was analyzed according to previous metabolic status and its association with mortality and post-acute COVID-19. Methods. A population-based observational retrospective study was conducted on a cohort of 110,726 patients aged 12 years or more who were diagnosed with COVID-19 infection between June 1st, 2021, and 28 February 2022 on the island of Gran Canaria, Spain. Results. In the 347 patients who died, the combination of advanced age, male sex, cancer, immunosuppressive therapy, coronary heart disease, elevated total cholesterol and reduced high-density lipoprotein cholesterol (HDL-C) was strongly predictive of mortality (*p* < 0.05). In the 555 patients who developed post-acute COVID-19, the persistence of symptoms was most frequent in women, older subjects and patients with obstructive sleep apnea syndrome, asthma, elevated fasting glucose levels or elevated total cholesterol (*p* < 0.05). A complete vaccination schedule was associated with lower mortality (incidence rate ratio (IRR) 0.5, 95%CI 0.39–0.64; *p* < 0.05) and post-acute COVID-19 (IRR 0.37, 95%CI 0.31–0.44; *p* < 0.05). Conclusions. Elevated HDL-C and elevated total cholesterol were significantly associated with COVID-19 mortality. Elevated fasting glucose levels and elevated total cholesterol were risk factors for the development of post-acute COVID-19.

## 1. Introduction

Since December 2019, coronavirus disease 2019 (COVID-19) has been an international public health emergency [1]. As of 22 June 2022, there have been more than 545 million cases and more than 6.3 million deaths worldwide [2]. The estimated rate of mortality among confirmed cases is 2.3% [3]. Initial studies showed that a worse prognosis was associated with advanced age, male sex and pre-existing comorbidities [4]. More recent studies have also demonstrated that obesity, diabetes and high blood pressure (hypertension) also increase the risk of severe COVID-19 [5,6,7]. Further studies evidenced that laboratory test results, such as blood levels of HDL-C or total cholesterol, may also have a predictive value in COVID-19 prognosis [8].

Obesity and its association with disease severity have also been reported for other respiratory viral infections; for example, 60% of patients who died of influenza A (H1N1 virus) were obese [9]. Moreover, patients with other comorbidities, such as alterations in total cholesterol or fasting glucose levels, were admitted to the intensive care unit more often and presented higher mortality [10]. However, the long-term consequences of obesity and previous metabolic status in patients with COVID-19 are not clear.

Multiple cardiovascular and metabolic risk factors have been shown to be associated with the risk of severe COVID-19 disease. At present, little is known about the association between post-acute COVID-19 and glycemic parameters and lipid levels [11].

The study hypothesis is that alterations in analytical determinations and patient comorbidities may play a role in COVID-19 infection. This study was aimed at evaluating this issue, as well as its relationship with post-acute COVID-19. In addition, the impacts of age, sex, vaccination, immunosuppressive treatment, patient comorbidities, and blood glucose or lipid levels and their association with developing post-acute COVID-19 or dying were analyzed.

## 2. Materials and Methods

Design: Population-based, observational retrospective cohort study.

Study area: A cohort of 110,726 patients who lived on the island of Gran Canaria, Spain.

Study group: A total of 110,726 patients aged 12 years or more who were diagnosed with COVID-19.

Eligibility criteria. The inclusion criteria were patients aged 12 years or more who were diagnosed with COVID-19 between 1 June 2021 and 28 February 2022 on the island of Gran Canaria. The exclusion criteria were as follows: age < 12 years.

Variables: Age, sex, personal history (asthma, cancer, dementia, diabetes, coronary heart disease, chronic obstructive pulmonary disease (COPD), high blood pressure, congestive heart failure (CHF), obesity, obstructive sleep apnea syndrome (OSAS), body mass index (BMI), date of first, second and booster doses of a COVID-19 vaccine, type of COVID-19 vaccine (Pfizer, Moderna, AstraZeneca or Janssen), death, immunosuppressive treatment, diagnosis of post-acute COVID-19 syndrome, laboratory test results (total cholesterol, low- and high-density lipoprotein cholesterol (LDL-C and HDL-C), triglycerides and fasting glucose levels), body weight, height and diagnostic test results (polymerase chain reaction with reverse transcription (RT-PCR), antigen test or serology).

Definitions: Participants were considered to suffer from diabetes if they had fasting glucose levels ≥ 126 mg/dL or were receiving antidiabetic therapy. They were considered to present obesity if BMI ≥ 30 kg/m^2^; in the age group 12 to 18 years, BMI was calculated in the same way as in adults, measuring height and weight. The BMI value and age of the participants in this group were then located on a sex-specific BMI-for-age table. This indicated whether these participants were in the range of obesity [12]. Participants were considered to suffer from hypertension if they had systolic blood pressure ≥ 140 mmHg and/or diastolic blood pressure ≥ 90 mmHg or received antihypertensive treatment. The participants were defined as known OSAS when there was a previous sleep study and/or the initiation of treatment documented by a physician. Regarding dyslipidemia, they were considered to present hypercholesterolemia if total cholesterol > 200 mg/dL; hypertriglyceridemia if triglycerides > 150 mg/dL; low HDL-C if HDL levels < 40 mg/dL (men) or <50 mg/dL (women); and high LDL-C if LDL levels > 130 mg/dL, or if they were receiving treatment for dyslipidemia [13].

Confirmed COVID-19 cases: Patients who met the clinical criteria for suspected COVID-19 and showed positive results in AIDT (active infection diagnostic test) or asymptomatic patients with positive AIDT plus a negative or not undertaken IgG-test. Suspected COVID-19 cases: Patients with acute respiratory infection of sudden onset of any degree of severity who presented with fever, cough or shortness of breath, among other signs. Further signs or symptoms such as odynophagia, anosmia, ageusia, muscle pain, diarrhea, chest pain, chills, fatigue, nausea and vomiting were also considered symptoms of suspected SARS-CoV-2 infection, depending on the doctor’s criterion.

Definition of post-acute COVID-19: A considerable number of patients experience persistent and debilitating symptoms after acute COVID-19 infection [14]. Greenhalgh et al. coined the expression “post-acute COVID-19 syndrome” for symptoms that persist for more than three weeks after the initial infection and cannot be explained by other causes [15,16], which is the criterion we followed in this study.

Complete vaccination schedule: Patients were considered to be fully vaccinated if (1) they had received 2 doses of the vaccine separated by a minimum of: 19 days if the first dose was BNT162b2 mRNA (Pfizer-BioNTech), 21 days if it was ChAdOx1 nCoV-19 (AstraZeneca-University of Oxford) or 25 days if it was mRNA-1273 (Moderna); and (2) if the minimum time elapsed since the last dose was: 7 days if the last dose was Pfizer or 14 days if it was AstraZeneca or Moderna. Patients were also considered to be fully vaccinated if they had received one dose of Ad26.COV2.S (Janssen) more than 14 days before. Patients up to 65 years old were also considered fully vaccinated if they had recovered from the disease and subsequently received a dose of any of the vaccines after the corresponding mentioned period for the second dose. Subjects vaccinated with a heterologous schedule consisting of a first dose of AstraZeneca and a second dose of an mRNA vaccine were considered fully vaccinated after 7 days if the second dose was Pfizer or 14 days if it was Moderna [17].

Data source and collection: The identification data of all patients who were vaccinated against COVID-19 in Gran Canaria (from 28 December 2020 to 28 February 2022) were obtained from REGVACU (the registry of vaccination against COVID-19 in Spain). The identification data of all COVID-19 cases in Gran Canaria that were notified to the Epidemiological Surveillance Network of the Canary Islands (REVECA) were obtained from the General Directorate of Public Health (DGSP) (period: 1 June 2021 to 28 February 2022). Post-vaccination COVID-19 cases reported to the DGSP were identified by combining both databases. The clinical information of patients diagnosed with COVID-19 was obtained from their primary care electronic medical records (DRAGO AP). DRAGO is the healthcare management system of the Canary Islands.

The information about the metabolic status of the subjects was obtained from measurements made during the dates selected for the study period, that is, between 1 June 2021 and 28 February 2022. After the end of the study period, the last available measurements of the analytical determinations (total cholesterol, triglycerides, HDL-C, LDL-C and fasting glucose levels) and the anthropometric parameters (abdominal perimeter, body weight, height and BMI) were selected.

Statistical analysis: A descriptive analysis of the results was carried out using frequency and percentages for categorical variables and mean and standard deviation (SD) for analytical determinations and quantitative variables. The cumulative incidence of post-acute COVID-19 was calculated as the ratio between the number of post-acute COVID-19 cases among people infected with SARS-CoV-2 in Gran Canaria (numerator) and the total number of people infected with SARS-CoV-2 (denominator) during the studied period. Bivariate analysis of qualitative variables was carried out with the χ2 test, using the Likelihood Ratio when necessary. In addition to the bivariate analysis, a multivariable Poisson regression model adjusting for predefined covariates was used to estimate the propensity scores for cohort participants. The models were used to determine the predictive values of death and post-acute COVID-19, which were defined as the dependent categorical variables in the analysis, adjusted by age, sex, immunosuppressive treatment, analytical determinations (LDL-C, HDL-C, total cholesterol, triglycerides and fasting glucose levels), type of COVID-19 vaccine, complete vaccination schedule, booster dose (3rd dose) by age and comorbidities, including diabetes, coronary heart disease, atrial fibrillation, hypertension, COPD, asthma, CHF, cancer, obesity, OSAS and dementia. Statistical significance was established at 5% (*p* < 0.05), and the level of confidence was set at 95%. Data were analyzed with the Statistical Package for the Social Sciences (SPSS) v20 and Microsoft^®^ Excel (2010).

## 3. Results

The study included 110,726 patients who were diagnosed with COVID-19 between June 2021 and February 2022. A total of 347 patients had died (0.3%). The mean age was 41 years (SD 16.8) and the predominant age group in the sample was 18–49 years; 55.3% of subjects were women. Mean BMI was 26.2 kg/m2 (SD 6.7): 27.2% of patients were overweight, 22.4% were obese and 3.5% were morbidly obese. The mean BMI of deceased patients was 29.5 kg/m^2^ (SD 6.3). The mean body weight in the sample was 70.5 kg (SD 22), while the mean height was 162 cm (SD 15.8). The mean abdominal perimeter was 92.1 cm (SD 19.3) for the total sample, 93.1 cm (SD 19.76) for men and 91.5 cm (SD 19.04) for women. Mean laboratory test results were: total cholesterol 183.3 mg/dL (SD 38.3), LDL-C 109.4 mg/dL (SD 32.7), HDL-C 55.2 mg/dL (SD 13.8), triglycerides 110.2 mg/dL (SD 61.6) and baseline glucose 98.1 mg/dL (SD 22.4).

The cumulative incidence of post-acute COVID-19 (number of cases per 1000 COVID-19 diagnosed subjects) was 5.01 in the total sample, 4.00 in subjects with complete vaccination and 9.71 in those who were not vaccinated. In obese subjects, the cumulative incidence was 5.30, while in non-obese individuals, it was 5.01.

Most cases of COVID-19 were diagnosed through PCR (44,284; 51%) or antigen test (41,942; 48.3%), followed by serology test (613; 0.7%). Each patient underwent a mean of 2.4 PCR tests (SD 1.7) and 1.3 antigen tests (SD 0.6).

The Pfizer vaccine was the most frequently applied as the first dose (62,714; 56.6%), followed by Moderna (20,972; 18.9%), AstraZeneca (8626; 7.8%) and Janssen (7549; 6.8%). For the second dose, Pfizer was again the most frequently used (52,601; 47.5%), followed by Moderna (30,500; 27.5%), AstraZeneca (8,065; 7.3%) and no second dose (19,560; 17.7%). As for the third dose, Moderna was the most frequently administered (11,269; 10.2%), followed by Pfizer (10,661; 9.6%) and no booster dose (88,796; 80.2%). A total of 10,865 patients (9.8%) had not received any dose.

The 555 (0.5%) patients who presented post-acute COVID-19 reported the following symptoms in order of frequency: asthenia (435; 74.4%), tiredness (154; 27.7%), fatigue (72; 13%), cognitive impairment (47; 8.5%), anosmia (31; 5.6%), malaise (28; 5%) and disorientation (12; 2.2%).

Patients with post-acute COVID-19 had a higher mean BMI than those without post-acute COVID-19 (26.4 kg/m^2^ vs. 26.2 kg/m^2^, respectively), although the difference was not significant (*p* = 0.953). Patients with post-acute COVID-19 were in turn classified into two sub-groups according to their BMI, the non-obese group (BMI < 30 kg/m^2^) or obese group (BMI ≥ 30 kg/m^2^), for further analysis.

Table 1 shows that the persistence of symptoms was more frequent in women, hypertensive, diabetic and older patients, and patients with coronary heart disease, CHF, cancer, OSAS or altered fasting glucose levels. A bivariate analysis of post-acute COVID-19 failed to show any statistically significant relationship with variables: cancer, immunosuppressive treatment, blood lipids and obesity (BMI > 30).

A bivariate analysis revealed a higher proportion of men among deceased patients. In addition, deceased patients were older and had more personal history issues (diabetes, cancer, hypertension, COPD, coronary heart disease, obesity, CHF or OSAS), immunosuppressive treatment, elevated triglycerides, elevated LDL-C, elevated total cholesterol, lower HDL-C, higher fasting glucose levels and incomplete COVID-19 vaccination (Table 2).

A multivariate analysis (Figure 1) showed that female sex, older age, OSAS, asthma, lack of complete vaccination, elevated fasting glucose levels and elevated total cholesterol were risk factors for the development of post-acute COVID-19 (*p* < 0.05).

A multivariate analysis showed that the combination of advanced age, male sex, cancer, coronary heart disease, elevated total cholesterol, immunosuppressive treatment, low HDL-C and lack of complete vaccination schedule or booster dose was strongly predictive of mortality (Figure 2). Mortality was negatively associated with a complete vaccination schedule in a risk factor analysis (incidence rate ratio 0.44, 95%CI 0.32–0.62; *p* < 0.05). The booster dose was associated with lower mortality (incidence rate ratio 0.29, 95%CI 0.22–0.39, *p* < 0.05).

## 4. Discussion

This study showed that high fasting glucose levels, low HDL-C and elevated total cholesterol were associated with a poorer COVID-19 prognosis and should be considered high-risk markers. Elevated fasting glucose levels and elevated total cholesterol were risk factors for the development of post-acute COVID-19. Other studies have already evaluated the association between alterations in analytical determinations and the patient’s previous comorbidities and their association with a worse COVID outcome (admission to the ICU, mechanical ventilation or death) [8]. However, until now, the association between lipid profile alterations and fasting glucose levels in the development of post-acute COVID-19 has not been evaluated.

Subjects with comorbidities and older subjects (who often present comorbidities) are especially vulnerable to acquiring acute COVID-19 infection and to meeting the criteria for severity during the acute phase, with consequent aftereffects for survivors. Increased morbidity and mortality in older patients and in patients with comorbidities have been associated with both comorbidities and frailty, which entail a poorer immune response [18].

Regarding COVID-19-associated comorbidities, more than half of the patients in our study presented one or more. Cascella et al. concluded that 49% of critical COVID-19 cases had pre-existing comorbidities, while Mughal et al. found that comorbidities such as obesity, diabetes or hypertension increased severity and mortality rates (10.5% with comorbidity vs. 0.9% without comorbidity) [19,20].

Immunosuppression was associated with mortality during the acute phase of COVID-19. However, no relationship was found with post-acute COVID-19 syndrome, in line with the results published by Carod-Artal et al. [21].

Hypertension, asthma and diabetes were the most frequent comorbidities in our study and were associated with higher mortality. However, they were not found to be independent factors associated with mortality in the multivariate analysis. The association between hypertension and worse outcomes of COVID-19 infection may be due to the higher frequency of comorbidities and the more advanced age of these individuals. An Italian cross-sectional study did not find hypertension as an independent factor affecting the outcome of COVID-19 [22].

In contrast, age, sex, coronary heart disease, hypercholesterolemia, cancer, immunosuppressive treatment, complete vaccination schedule, booster dose and low HDL-C levels were independently associated with mortality, although they affected only a small proportion of patients.

It has been demonstrated that optimal body weight/composition is a protective factor against different diseases, such as COVID-19. In this context, Cao et al. found lower mortality rates in subjects with BMIs within the “normal” range [23]. However, the results concerning obesity are controversial. Dietz et al. reported that most of the patients admitted to the intensive care unit for COVID-19 in Italy were obese, a finding that was associated with the high mortality rates recorded in Italy vs. countries with lower obesity rates, such as China [24]. However, in our sample, post-acute COVID-19 was not more frequent among subjects with obesity or higher BMIs than in the others.

It is important to highlight that HDL-C is part of the innate immune system and may neutralize toxic materials produced by infection and inflammation processes, thus moderating the inflammatory burst. This protective mechanism could be impaired in COVID-19 diagnosed patients with low HDL-C. Thus, low HDL levels could not only be a side effect of inflammation and infection but also contribute to a poorer outcome of the disease [8]. These findings support the use of these serum markers to assess the progression of the disease, as well as the need to perform routine laboratory tests in the management of COVID-19 patients.

Kuderer N.M. et al. reported an association between cancer and higher mortality rates due to COVID-19. The immune system of cancer patients may be weakened by the effects of antineoplastic therapy and supportive drugs such as steroids and the immunosuppressive effects of cancer itself [25]. In our study, the overall prevalence of active cancer as a comorbidity was 3.2%, and it was an independent factor associated with mortality in the multivariate analysis. Other authors such as Gu et al. provided substantial statistical evidence for the value of coronary heart disease as a predictor of COVID-19 mortality [26].

There may also be protective factors for the post-COVID-19 period; a recent study suggested that vaccines may offer such protection [27]. In our study, receiving two doses of the COVID-19 vaccine was associated with a substantial decrease in the most common post-COVID-19 symptoms and lower mortality. Subjects who had received two doses reported such symptoms less frequently than those who were not vaccinated [28]. Arben et al. found that patients who had received a booster dose at least 5 months after a second dose of Pfizer-BioNTech had 90% lower mortality due to COVID-19 than patients who had not received a booster [29].

Risk factors for the development of post-acute COVID-19 identified in our study included: age (older than 50 years), female sex and pre-existing comorbidities, such as asthma. Similar findings were reported by Chen et al., where female patients and patients with pre-existing asthma were more likely to develop a post-COVID-19 condition, with estimated pooled probability ratios (OR) of 1.57 (95%CI 1.09, 2.26) and 2.15 (95%CI 1.14, 4.05), respectively [30].

The proportion of hyperglycemia among patients diagnosed with post-acute COVID-19 in our study was surprisingly high, which leads us to conclude that more aggressive glycemic control is required [31]. A two-way interaction between COVID-19 and diabetes has been recently evidenced, which involves a complex physiopathological characteristic underlying hyperglycemia and generally impaired glycometabolism. Montefusco et al. demonstrated new-onset hyperglycemia, insulin resistance and beta-cell hyperstimulation in COVID-19 patients with no history of diabetes; such effects could be observed long after remission [32]. Moreover, recent studies by Xie et al. found higher risks of developing cardiovascular disease and diabetes in COVID-19 patients after one year [33,34].

The inflammatory response to SARS-CoV-2 infection increases the production of hyperglycemic hormones such as cortisol and those related to adrenergic discharge [35]. It has been postulated that hyperglycemia is a determining element in certain cardiovascular complications due to its involvement in platelet and monocyte activation, which might lead to increased SARS-CoV-2 virulence [36].

OSAS had a considerable impact on the progression of COVID-19. The development of post-acute COVID-19 in patients with OSAS was higher in the multivariable Poisson regression model, independently of other variables (incidence rate ratio = 1.840). Thus, OSAS should be considered a significant risk factor in post-acute COVID-19 risk factor analyses [37].

There are certain epidemiological limitations to be considered when interpreting the data: the heterogeneity of samples in terms of age (younger vs. older age groups), different definitions of post-acute COVID-19 syndrome or variations in the inclusion criteria, as defined by the persistence of at least one clinically relevant symptom. The lack of a standardized and generally accepted definition hinders the comparability of epidemiological studies. In addition, virus variants and pandemic periods (waves) were not considered in the analysis. One of the weak points of this study is the high prevalence of overweight in our population; this made the sample rather heterogeneous and resulted in a large BMI dispersion (26.2 kg/m2, SD 6.7), which may have led to contradictory results. It is possible that the glycemic and lipidemic parameters are elevated in obese or overweight subjects. There is a phenotype corresponding to individuals who have a normal weight but are metabolically obese; that is, they have a normal BMI but present the typical alterations of obese patients: insulin resistance, low levels of HDL-C and high concentrations of triglycerides. At the same time, there are those who have been called metabolically healthy obese. These individuals have a BMI > 30 but none of the metabolic abnormalities typical of obese individuals [38].

The main strong point of the study is its large number of participants, much larger than most of the Spanish studies on this subject. Furthermore, objective measurements were used for anthropometric parameters (abdominal perimeter, body weight, height and BMI) and laboratory tests (total cholesterol, triglycerides, HDL-C, LDL-C and fasting glucose levels), which prevents underestimation biases derived from evaluating the prevalence of certain factors through questionnaires. These measurements were not used very often in earlier studies of SARS-CoV-2 patients but are helpful for defining metabolic syndrome more specifically. Further strong points were the inclusion of a control group of subjects without post-acute COVID-19, the fact that the study was not conducted during the initial phase of the pandemic and the inclusion of immunosuppressive therapy.

In conclusion, this study confirmed a link between age, sex, comorbidities, immunosuppressive treatment, and vaccination and dying from COVID-19. Obesity was not demonstrated to be a risk factor for the development of post-acute COVID-19. However, post-acute COVID-19 was associated with age, sex, asthma, OSAS, no complete vaccination schedule, elevated total cholesterol and altered fasting glucose levels. Blood levels of lipids may be useful predictors of mortality in COVID-19 patients. High fasting glucose levels, low HDL-C and elevated total cholesterol were associated with a poorer COVID-19 prognosis and should be considered high-risk markers. As shown in the present study, plasma lipid concentrations and fasting glucose levels should assist with the clinical management of COVID-19. This difference in disease course can help to assess patients’ prognosis and the occurrence of severe forms and thus provide optimal management. Two doses of the SARS-CoV-2 vaccine were highly effective in preventing COVID-19-related deaths in all age groups. More complete vaccination was associated with less frequent development of post-acute COVID-19. These findings suggest that COVID-19 vaccination can help in bringing the pandemic under control.

## Figures and Tables

**Figure 1 nutrients-14-02925-f001:**
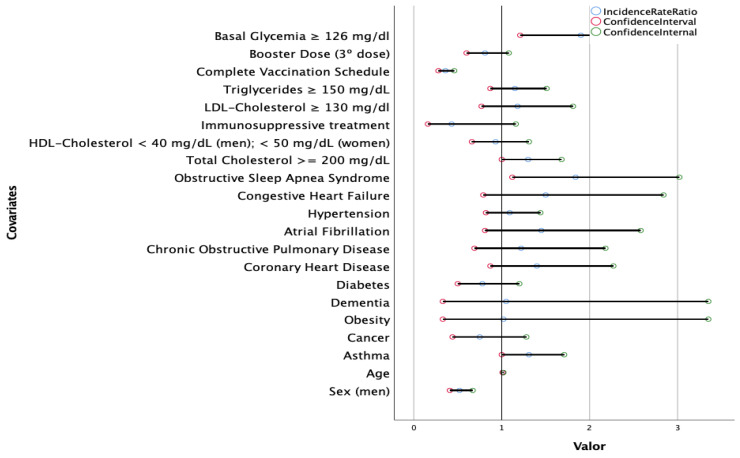
A multivariate Poisson regression model to explore the association between post-acute COVID-19 according to sex, age, comorbidities, laboratory test results, immunosuppressive treatment and vaccination schedule in 110,726 SARS-CoV-2-positive patients admitted to hospital. A multivariate Poisson regression model was used. *p* values of the variables of interest according to a variable permutation analysis: sex: <0.001; age: 0.008; asthma: 0.50; cancer: 0.296; obesity: 0.956; dementia: 0.938; diabetes: 0.258; coronary heart disease: 0.166; chronic obstructive pulmonary disease: 0.498; atrial fibrillation: 0.212; hypertension: 0.552; congestive heart failure: 0.218; obstructive sleep apnea syndrome: 0.017; total cholesterol: 0.044; HDL-cholesterol: 0.690; immunosuppressive treatment: 0.096; LDL-cholesterol: 0.450; triglycerides: 0.330; complete vaccination schedule: <0.001; booster dose: 0.142; fasting glucose levels: 0.005.

**Figure 2 nutrients-14-02925-f002:**
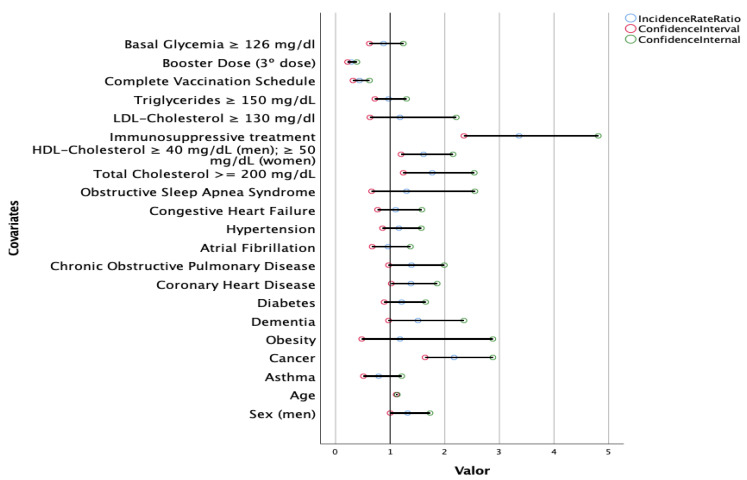
A multivariate Poisson regression model to explore the association between death according to sex, age, comorbidities, laboratory test results, immunosuppressive treatment and vaccination schedule in 110,726 SARS-CoV-2-positive patients admitted to hospital. *p* values of the variables of interest according to a variable permutation analysis: sex: 0.042; age: <0.001; asthma: 0.271; cancer: <0.001; obesity: 0.720; dementia: 0.069; diabetes: 0.230; coronary heart disease: 0.036; chronic obstructive pulmonary disease: 0.071; atrial fibrillation: 0.829; hypertension: 0.327; congestive heart failure: 0.587; obstructive sleep apnea syndrome: 0.447; total cholesterol: 0.002; HDL-cholesterol: 0.002; immunosuppressive treatment: <0.001; LDL-cholesterol: 0.609; triglycerides: 0.818; complete vaccination schedule: <0.001; booster dose: <0.001; fasting glucose levels: 0.450.

**Table 1 nutrients-14-02925-t001:** Association between predictor variables and post-acute COVID-19 in 110,726 participants.

Variables		N Cases (%)	N Post-Acute COVID-19	Odds Ratio (CI)	*p* Value
Sex	Women	61184 (55.3)	391	1.9 (1.6–2.3)	<0.001
	Men	49542 (44.7)	164	1	
Age	12–17	7.500 (6.8)	36	2.10 (1.40–3.14)	<0.001
	18–49	71.527 (64.6)	301	1	
	50–69	24.608 (22.2)	147	1.68 (1.27–2.24)	
	>70	7.091 (6.4)	71	2.39 (1.85–3.10)	
Immunosuppressive treatment	Yes	2057 (1.9)	7	0.67 (0.32–1.42)	0.297
	No	108,669 (98.1)	548	1	
Diabetes	Yes	7684 (6.9)	60	1.6 (1.2–2.1)	<0.001
	No	103,042 (93.1)	495	1	
Coronary heart disease	Yes	2525 (2.3)	25	2.0 (1.4–3)	<0.001
	No	108,201 (97.7)	530	1	
Atrial fibrillation	Yes	1515 (1.4)	18	2.4 (1.5–3.9)	<0.001
	No	109,211 (98.6)	537	1	
Hypertension	Yes	20,109 (18.2)	135	1.5 (1.2–1.8)	<0.001
	No	90,617 (81.8)	420	1	
Chronic obstructive pulmonary disease	Yes	1648 (1.5)	15	1.85 (1.1–3.09)	0.018
	No	109,078 (98.5)	540	1	
Asthma	Yes	15.295 (13.8)	96	1.31 (1.05–1.63)	0.017
	No	95.431 (86.2)	459	1	
Congestive heart failure	Yes	982 (0.9)	15	3.1 (1.9–5.3)	<0.001
	No	109,744 (99.1)	540	1	
Cancer	Yes	3579 (3.2)	24	1.4 (0.9–2)	0.145
	No	107,147 (96.8)	531	1	
Obesity	Yes	1513 (1.4)	8	1.06 (0.52–2.13)	0.879
	No	109,213 (98.6)	547	1	
Obstructive sleep apnea syndrome	Yes	2154 (1.9)	20	1.9 (1.2–3)	0.005
	No	108,037 (98.1)	535	1	
Fasting glucose levels	≥126 mg/dL	4333 (5.7)	43	1.9 (1.4–2.6)	<0.001
	<126 mg/dL	71,274 (94.3)	374	1	
LDL cholesterol	<130 mg/dL	73,494 (93)	435	0.92 (0.63–1.33)	0.643
	≥130 mg/dL	5530 (7)	30	1	
HDL cholesterol	≥40 mg/dL (men); ≥50 mg/dL (women)	71,858 (87.3)	427	1.13 (0.85–1.5)	0.392
	<40 mg/dL (men); <50 mg/dL (women)	10,460 (12.7)	55	1	
Total cholesterol	<200 mg/dL	66,304 (69.6)	372	0.93 (0.77–1.13)	0.457
	≥200 mg/dL	28,909 (30.4)	151	1	
Triglycerides	<150 mg/dL	73,322 (81.8)	388	1	0.179
	≥150 mg/dL	16,261 (18.2)	100	1.16 (0.93–1.45)	
Type of COVID-19 vaccine	Pfizer	52.601 (47.5)	195	0.27 (0.22–0.34)	<0.001
	Moderna	30.500 (27.5)	129	0.31 (0.25–0.39)	
	AstraZeneca	8.065 (7.3)	41	0.38 (0.27–0.53)	
	Janssen	7.549 (6.8)	29	0.28 (0.19–0.42)	
	Not vaccinated	12.011 (10.9)	161	1	
Complete vaccination schedule	Yes	94,167 (85)	376	0.37 (0.31–0.44)	<0.001
	No	16,559 (15)	179	1	
Booster dose (3rd dose) 12–17 years	Yes	14 (0.2)	0	1	0.795
	No	7486 (99.8)	36	1.005 (1.003–1.006)	
Booster dose (3rd dose) 18–49 years	Yes	8090 (11.3)	34	1.001 (0.7–1.43)	0.994
	No	63,437 (88.7)	267	1	
Booster dose (3rd dose) 50–69 years	Yes	8887 (36.1)	42	1.42 (0.99–2.03)	0.056
	No	15721 (63.9)	105	1	
Booster dose (3rd dose) ≥70 years	Yes	4939 (69.6)	38	1	0.003
	No	2152 (30.4)	33	2 (1.26–3.21)	

**Table 2 nutrients-14-02925-t002:** Association between predictor variables and death in 110,726 participants.

Variables		N Cases	Deaths	Odds Ratio (CI)	*p* Value
Sex	Women	61,184 (55.3)	161	0.7 (0.6–0.9)	<0.001
	Men	49,542 (44.7)	186	1	
Age	12–49	79.027	11	1	<0.001
	50–69	24.608	69	20.2 (10.69–38.17)	
	≥70	7.091	267	281.1 (153.71–513.93)	
Immunosuppressive treatment	Yes	2057 (1.9)	50	9.1 (6.7–12.3)	<0.001
	No	108,669 (98.1)	297	1	
Diabetes	Yes	7684 (6.9)	148	10.1 (8.2–12.6)	<0.001
	No	103,042 (93.1)	199	1	
Coronary heart disease	Yes	2525 (2.3)	89	15.3 (12–19.5)	<0.001
	No	108,201 (97.7)	258	1	
Atrial fibrillation	Yes	1515 (1.4)	64	17 (12.8–22.4)	<0.001
	No	109,211 (98.6)	283	1	
Hypertension	Yes	20,109 (18.2)	242	10.5 (8.3–13.2)	<0.001
	No	90,617 (81.8)	105	1	
Chronic obstructive pulmonary disease	Yes	1648 (1.5)	53	12.3 (9.1–16.5)	<0.001
	No	109,078 (98.5)	1648	1	
Asthma	Yes	15.295 (13.8)	33	0.66 (0.46–0.94)	0.020
	No	95.431 (86.2)	314	1	
Congestive heart failure	Yes	982 (0.9)	65	27.5 (20.8–36.3)	<0.001
	No	109,744 (99.1)	282	1	
Cancer	Yes	3579 (3.2)	103	13 (10.3–16.4)	0.005
	No	107,147 (96.8)	244	1	
Obesity	Yes	1513 (1.4)	6	1.27 (0.57–2.85)	0.560
	No	109,213 (98.6)	341	1	
Obstructive sleep apnea syndrome	Yes	2154 (1.9)	14	2.1 (1.2–3.6)	0.005
	No	108,037 (98.1)	333	1	
Fasting glucose levels	≥126 mg/dL	4333 (5.7)	83	6.2 (4.8–8)	<0.001
	<126 mg/dL	71,274 (94.3)	223	1	
LDL cholesterol	<130 mg/dL	73,494 (93)	281	0.57 (0.32–1)	0.051
	≥130 mg/dL	5530 (7)	12	1	
HDL cholesterol	≥40 mg/dL (men); ≥50 mg/dL (women)	71,858 (87.3)	100	0.3 (0.24–0.38)	<0.001
	<40 mg/dL (men); <50 mg/dL (women)	10,460 (12.7)	208	1	
Total cholesterol	<200 mg/dL	66,304 (69.6)	263	0.53 (0.4–0.7)	<0.001
	≥200 mg/dL	28,909 (30.4)	61	1	
Triglycerides	<150 mg/dL	73,322 (81.8)	228	1	<0.001
	≥150 mg/dL	16,261 (18.2)	99	2.1 (1.5–2.8)	
Type of COVID-19 vaccine	Pfizer	52.601 (47.5)	125	0.32 (0.25–0.42)	<0.001
	Moderna	30.500 (27.5)	111	0.49 (0.37–0.66)	
	AstraZeneca	8.065 (7.3)	16	0.27 (0.16–0.46)	
	Janssen	7.549 (6.8)	7	0.13 (0.06–0.27)	
	Not vaccinated	12.011 (10.9)	88	1	
Complete vaccination schedule	Yes	94,167 (85)	257	0.5 (0.39–0.64)	<0.001
	No	16,559 (15)	90	1	
Booster dose (3rd dose) 12–17 years	Yes	14 (0.2)	0	1	
	No	7486 (99.8)	0		
Booster dose (3rd dose) 18–49 years	Yes	8090 (11.3)	2	1	0.472
	No	63,437 (88.7)	9	0.574 (0.124–2.66)	
Booster dose (3rd dose) 50–69 years	Yes	8887 (36.1)	18	1	0.082
	No	15,721 (63.9)	51	1.6 (0.94–2.75)	
Booster dose (3rd dose) ≥70 years	Yes	4939 (69.6)	94	1	<0.001
	No	2152 (30.4)	173	4.51 (3.5–5.8)	

## Data Availability

The data are not publicly available due to privacy or ethical reasons. Data are available from the management of Primary Care of Gran Canaria, Spain, for researchers who meet the criteria for access to confidential data.

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
