# Peer review of "Assessment of SARS-CoV-2 Infection According to Previous Metabolic Status and Its Association with Mortality and Post-Acute COVID-19"

_nutrients, 2022, doi:10.3390/nu14142925_

Round 1
Reviewer 1 Report
Thank you for the opportunity to review your manuscript. The topic of the paper seems relevant from the point of view that the issue of COVID-19 is still relevant in the public space, and scientific papers on this disease continue to appear in the pages of many journals.
Despite the significant scientific contribution, the authors failed to avoid several issues.
First, the introduction is too short to adequately outline the background of the research problem. In addition, I recommend that the research objective be a separate paragraph and cite the research questions or hypotheses that the Authors confirmed/disconfirmed. I recommend that the methodology section be divided into appropriate sections: study area, study group, eligibility criteria, research instrument, ethical consents, statistical analyses. And in relation to these sections, I have a question, where in the paper are the eligibility criteria? I understand that this is retrospective data, but I think the selection criteria should be clearly defined. Did the authors consider virus variants and pandemic periods (waves) in their analyses? If not then I suggest including this in the limitations of the study.
Kind regards!
Author Response
Dear Mr. Reviewer,
Regarding the manuscript: "Assessment of SARS-CoV-2 infection according to previous metabolic status and its association with mortality and post-acute COVID-19" (Manuscript ID: nutrients-1815494)
Thank you very much for your feedback and for the opportunity to resubmit a second revised version for publication. We have addressed all issues and thank the reviewers for their constructive criticism.
We hope that the manuscript is now suitable for publication.
We look forward to hearing from you,
Sincerely,
Corresponding author, on behalf of all authors.
COMMENTS FOR REVIEWERS. MODIFICATIONS.
We thank the reviewers for their constructive comments that certainly improve the manuscript. The changes we have made to the manuscript are highlighted in yellow.
Point 1: First, the introduction is too short to adequately outline the background of the research problem.
Response 1: We thank the reviewer for this suggestion. We have further developed the introduction according to the research question.
Multiple cardiovascular and metabolic risk factors have been shown to be associated with the risk of severe COVID-19 disease. So far, little is known about the association between post-acute COVID-19 and the glycemic parameters and lipid levels [11].
Del Sole, F.; Farcomeni, A.; Loffredo, L.; Carnevale, R.; Menichelli, D.; Vicario, T.; Pignatelli, P.; Pastori, D. Features of severe COVID-19: A systematic review and meta-analysis. Eur. J. Clin. Investig. 2020, 50, e13378.
We have made these changes to the text. Please see the Introduction section (page 2, lines 45-47).
Point 2: In addition, I recommend that the research objective be a separate paragraph and cite the research questions or hypotheses that the Authors confirmed/disconfirmed.
Response 2: We thank the reviewer for this suggestion.
We have separated the paragraph in which we refer to the objectives of the study and cite the research questions or hypotheses that the Authors confirmed/disconfirmed.
The study hypothesis is that alterations in analytical determinations (glucose and cholesterol) and patient comorbidities may play a role in COVID-19 infection.
We have made these changes to the text. Please see the Introduction section (page 2, lines 48-50).
Point 3: I recommend that the methodology section be divided into appropriate sections: study area, study group, eligibility criteria, research instrument, ethical consents, statistical analyses.
And in relation to these sections, I have a question, where in the paper are the eligibility criteria? I understand that this is retrospective data, but I think the selection criteria should be clearly defined.
Response 3: We thank the reviewer for this suggestion.
We have divided the methodology section into appropriate sections: design, study area, study group, eligibility criteria, definitions, variables, data source and collection, ethical consents, informed consent statement and statistical analyses.
Exclusion criteria was as follows: age < 12 years.
Please see the Materials and Methods section (page 3, line 62)
Point 4: Did the authors consider virus variants and pandemic periods (waves) in their analyses? If not, then I suggest including this in the limitations of the study.
Response 4: We thank the reviewer for this suggestion. In this study, virus variants and pandemic periods (waves) were not considered for analysis.
Please see the Discussion section (page 10, lines 320-321).
Please see the attachment

Reviewer 2 Report
Thank you so much for inviting me to review this manuscript. The article is very interesting in terms of subject matter. However, some methodological aspects should be improved. I have some comments to improve the manuscript.
- Since this is a cross-sectional study, authors cannot establish cause-effect relationships. Some terms (e.g., "risk", "effects") are misleading with this study design.
- I would exclude minors from testing. What criteria were used for them in terms of BMI? This needs to be stated in the manuscript.
- Why did the authors use logistic regression analysis instead of Poisson regression? Logistic regressions tend to overestimate the coefficients of the ORs. I suggest modifying the analyses, given the relevance of the proposed topic.
- In line 113, do the authors refer to covariates instead of confounders? These are different terms.
- Please, replace "p=0.000" with "p<0.001" (e.g. Tables).
Kind regards,
Author Response
Dear Mr. Reviewer,
Regarding the manuscript: "Assessment of SARS-CoV-2 infection according to previous metabolic status and its association with mortality and post-acute COVID-19" (Manuscript ID: nutrients-1815494)
Thank you very much for your feedback and for the opportunity to resubmit a second revised version for publication. We have addressed all issues and thank the reviewers for their constructive criticism.
We hope that the manuscript is now suitable for publication.
We look forward to hearing from you,
Sincerely,
Corresponding author, on behalf of all authors.
Thank you so much for inviting me to review this manuscript. The article is very interesting in terms of subject matter. However, some methodological aspects should be improved. I have some comments to improve the manuscript.
COMMENTS FOR REVIEWERS. MODIFICATIONS.
We thank the reviewers for their constructive comments that certainly improve the manuscript. The changes we have made to the manuscript are highlighted in yellow.
Point 1: Since this is a cross-sectional study, authors cannot establish cause-effect relationships. Some terms (e.g., "risk", "effects") are misleading with this study design.
Response 1: We thank the reviewer for this contribution. We have replaced these terms with others according to the design of the study.
The terms of risk and effects that remain in the manuscript refer to other articles cited in the bibliography.
Point 2: I would exclude minors from testing. What criteria were used for them in terms of BMI? This needs to be stated in the manuscript.
Response 2: We thank the reviewer for this contribution. In the age group 12 to 18 years, BMI was calculated in the same way as in adults, measuring height and weight. The BMI number and age of the participants in this group were then located on a sex-specific BMI-for-age table. This indicated whether these participants were in the range of obesity.
Please see the Materials and Methods section (page 2, lines 65-68).
Point 3: Why did the authors use logistic regression analysis instead of Poisson regression? Logistic regressions tend to overestimate the coefficients of the ORs. I suggest modifying the analyses, given the relevance of the proposed topic.
Response 3: We thank the reviewer for this suggestion.
We have provided more details on how logistic regression was performed.
Multinomial logistic regression adjusting for predefined covariates was used to estimate the propensity scores for cohort participants.
Multivariable logistic regression was used to model the relationships between risk factors and clinical outcomes.
The multivariate binary logistic regression models were used to determine the predictive value of death and post-acute COVID-19, which was defined as an independent categorical variable in the analysis, adjusted by age, sex, immunosuppressive treatment, analytical determinations (LDL-C, HDL-C, total cholesterol, triglycerides and basal glycemia), type of COVID-19 vaccine, complete vaccination schedule, booster dose (3º dose) by age and comorbidities including diabetes, coronary heart disease, atrial fibrillation, hypertension, chronic obstructive pulmonary disease, asthma, congestive heart failure, cancer, obesity, obstructive sleep apnea syndrome, dementia).
Please see the Materials and Methods section (page 2, lines 135-146).
In addition, several of the studies that we reference throughout our manuscript and with which we want to compare ourselves in the discussion used logistic regression as the method of analysis. Here are some examples:
Multivariate logistic regression was modeled following stepwise selection to choose the best predictors. Odds ratio with 95% confidence intervals were calculated.
Mughal, M. S.; Kaur, I. P.; Jaffery, A. R.; Dalmacion, D. L.; Wang, C.; Koyoda, S.; Kramer, V. E.; Patton, C. D.; Weiner, S.; Eng, M. H.; Granet, K. M. COVID-19 patients in a tertiary US hospital: Assessment of clinical course and predictors of the disease severity. Respir Med 2020, 172, 106130.
Univariate and multivariate binary logistic regression models were used to determine the predictive value of OSA, which was defined as an independent categorical variable in the analysis, adjusted by age, sex, Body Mass Index (BMI; entered as a continuous variable unless otherwise specified), and comorbidities including diabetes mellitus, cardiovascular disease, and obstructive lung disease, for prediction of adverse clinical outcomes of COVID-19 including mortality, ICU admission, and hospital stay > 1 week.
Voncken, S.F.J.; Feron, T.M.H.; Laven, S.A.J.S.; Karaca, U.; Beerhorst, K.; Klarenbeek, P.; Straetmans, J.M.J.A.A.; de Vries, G.J.; Kolfoort-Otte, A.A.B.; de Kruif, M.D. Impact of obstructive sleep apnea on clinical outcomes in patients hospitalized with COVID-19. Sleep Breath. 2021, 24, 1–9.
The propensity score, a predicted probability of glucose change contributed by the above variables, were estimated based on multivariable logistic regression model.
Zhu, L.; She, Z.-G.; Cheng, X.; Qin, J.-J.; Zhang, X.-J.; Cai, J.; Lei, F.; Wang, H.; Xie, J.; Wang, W.; et al. Association of Blood Glucose Control and Outcomes in Patients With COVID-19 and Pre-Existing Type 2 Diabetes. Cell Metab. 2020, 31, 1068–1077.e3.
Multinomial logistic regression adjusting for predefined covariates was used to estimate the propensity scores for cohort participants.
Xie, Y.; Al-Aly, Z. Risks and burdens of incident diabetes in long COVID: A cohort study. Lancet Diabetes Endocrinol. 2022, 10, 311–321.
Multivariable logistic regression was used to model the relationships between risk factors and clinical outcomes
Montefusco, L.; Ben Nasr, M.; D’Addio, F.; Loretelli, C.; Rossi, A.; Pastore, I.; Daniele, G.; Abdelsalam, A.; Maestroni, A.; Dell’Acqua, M.; et al. Acute and long-term disruption of glycometabolic control after SARS-CoV-2 infection. Nat. Metab. 2021, 3, 774–785.
Point 4: In line 113, do the authors refer to covariates instead of confounders? These are different terms.
Response 4: We thank the reviewer for this contribution. We wanted to refer to the term covariates.
Please see the Materials and Methods section (page 3, lines 136).
Point 5: Please, replace "p=0.000" with "p<0.001" (e.g. Tables).
Response 5: We thank the reviewer for this contribution. We have replaced "p=0.000" with "p<0.001".
Please see the Results section (tables).
Please see the attachment

Reviewer 3 Report
Were there any exclusion criteria from the study?
Please provide more details on how logistic regression was performed.
How did the authors validate the model?
Did the authors assess sensitivity, specificity and ability to generalize?
It is recommended to present the results of logistic regression analysis in the forest plot.
It would be better to present the descriptive statistics of the studied population in the table.
In the Results section, the authors mentioned that the cumulative incidence of post-acute COVID-19 was 5.01 in the total sample. In the next sentence, the researchers reported that in obese subjects, the cumulative incidence was 5.71, while in non-obese, it was 5.74. Is there a mistake in this sentence?
In the tables, please change p=0.000 to p<0.001.
What is the novelty of the study?
Author Response
Dear Mr. Reviewer,
Regarding the manuscript: "Assessment of SARS-CoV-2 infection according to previous metabolic status and its association with mortality and post-acute COVID-19" (Manuscript ID: nutrients-1815494)
Thank you very much for your feedback and for the opportunity to resubmit a second revised version for publication. We have addressed all issues and thank the reviewers for their constructive criticism.
We hope that the manuscript is now suitable for publication.
We look forward to hearing from you,
Sincerely,
Corresponding author, on behalf of all authors.
COMMENTS FOR REVIEWERS. MODIFICATIONS.
We thank the reviewers for their constructive comments that certainly improve the manuscript. The changes we have made to the manuscript are highlighted in yellow.
Point 1: Were there any exclusion criteria from the study?
Response 1: We thank the reviewer for this suggestion. Exclusion criteria was as follows: age < 12 years.
Please see the Materials and Methods section (page 2, line 62).
Point 2: Please provide more details on how logistic regression was performed.
Response 2: We thank the reviewer for this contribution.
Multinomial logistic regression adjusting for predefined covariates was used to estimate the propensity scores for cohort participants.
Multivariable logistic regression was used to model the relationships between risk factors and clinical outcomes.
The multivariate binary logistic regression models were used to determine the predictive value of death and post-acute COVID-19, which was defined as an independent categorical variable in the analysis, adjusted by age, sex, immunosuppressive treatment, analytical determinations (LDL-C, HDL-C, total cholesterol, triglycerides and basal glycemia), type of COVID-19 vaccine, complete vaccination schedule, booster dose (3º dose) by age and comorbidities including diabetes, coronary heart disease, atrial fibrillation, hypertension, chronic obstructive pulmonary disease, asthma, congestive heart failure, cancer, obesity, obstructive sleep apnea syndrome, dementia).
Please see the Materials and Methods section (page 2, lines 135-146).
Point 3: How did the authors validate the model?
Response 3: We thank the reviewer for this contribution.
The model was validated through the Hosmer-Lemeshow and Nagelkerke (R2) tests. The Hosmer-Lemeshow test showed acceptable calibration for both models.
Death and multivariate analysis adjusted (95%CI): Nagelkerke (R2) tests: 0,356
Hosmer-Lemeshow test: 0,913
Post-acute COVID-19 and multivariate analysis adjusted (95%CI): Nagelkerke (R2) tests: 0,352. Hosmer-Lemeshow test: 0,480
Please see the Materials and Methods section (page 2, lines 144-146).
Point 4: It is recommended to present the results of logistic regression analysis in the forest plot.
Response 4: We thank the reviewer for this suggestion.
Please see the attachment
Point 5: It would be better to present the descriptive statistics of the studied population in the table.
Response 5: We thank the reviewer for this contribution. We have presented the descriptive statistics of the studied population in the table. However, the values that refer to means and standard deviation have been kept in the text because they will not be used for the bivariate analysis.
Point 6: In the Results section, the authors mentioned that the cumulative incidence of post-acute COVID-19 was 5.01 in the total sample. In the next sentence, the researchers reported that in obese subjects, the cumulative incidence was 5.71, while in non-obese, it was 5.74. Is there a mistake in this sentence?
Response 6: We thank the reviewer for this contribution.
The cumulative incidence of post-acute COVID-19 was 5.01 in the total sample.
In obese subjects, the cumulative incidence was 5.30, while in non-obese, it was 5.01. Please see the Results section (page 4, lines 162 and 164).
Point 7: In the tables, please change p=0.000 to p<0.001.
Response 7: We thank the reviewer for this contribution. We have replaced "p=0.000" with "p<0.001". Please see the Results section (tables).
Point 8: What is the novelty of the study?
Response 8: We thank the reviewer for this contribution.
The elevated basal glycemia was associated with higher risk of post-acute COVID-19.
High basal glycemia, low HDL-C and elevated total cholesterol were associated with a poorer COVID-19 prognosis and should be considered high-risk markers.
This study showed how elevated baseline glycemia was risk factors for the development of post-acute COVID-19 (p < 0.05).
Other studies have already evaluated the association between alterations in analytical determinations and the patient's previous comorbidities and their association with a worse COVID outcome (admission to the ICU, mechanical ventilation, or death). However, until now the association between lipid profile alterations and basal glycemia in the development of post-acute covid had not been evaluated.
Please see the first paragraph of the Discussion section (page 8, lines 228-235). discussion The best way to start any discussion is to present their own novelty research.
Please see the attachment

Reviewer 4 Report
Reviewer comments and suggestions
The authors of this study evaluated the SARS-CoV-2 infection based on the previous metabolic status and further found its association with mortality and post-acute COVID-19.
The study included a population-based observational retrospective study conducted on a cohort of 110,726 patients aged 12 years or more, who were diagnosed with COVID-19 infection between June 1st 2021 and February 28th 2022 in the island of Gran Canaria. The authors reported in 347 patients who died, the combination of several confounding and risk factors was strongly predictive of mortality (p < 0.05).
The important part of the result obtained in 555 patients who developed post-acute COVID-19, the persistence of symptoms was most frequent in women, older subjects, and patients with obstructive sleep apnea, asthma or elevated basal glycemia (p < 0.05). The study concluded that elevated basal glycemia, low HDL-C, and high total cholesterol should be added to the list of well-known risk factors for severe COVID-19.
For the improvement of the manuscript, I am suggesting a few comments.
- Line 34 please include more references for the line
- Line 36-37 the author's study was also relating these variables so what is the novelty of this study here already they reported.
- Line 59 SARS COV- 2 better presents the full form
- Line 80 In material and method the authors should write about the diagnostic approach to defining OSAS,
- In the material and method section, Patient consent has been taken or not, it should be written in the manuscript
- Line 97-101 need to mention the appropriate reference
- Table 3 multivariate analysis please specify the factors in legends
- First para of discussion The best way to start any discussion is to present their own novelty research rather than discussing others, please modify
- Line 197-198 is there any reason for this
- Line 236 Please cite those several meta-analyses
- Line 264-265 It is possible that the glycemic and lipidemic parameters are elevated in obese or overweight subjects, The authors need to check with different population's results and discuss.
- Line 284 What do the authors want to relate with COVID here
- All references need to be modified based on the MDPI journals.
Author Response
Dear Mr. Reviewer,
Regarding the manuscript: "Assessment of SARS-CoV-2 infection according to previous metabolic status and its association with mortality and post-acute COVID-19" (Manuscript ID: nutrients-1815494)
Thank you very much for your feedback and for the opportunity to resubmit a second revised version for publication. We have addressed all issues and thank the reviewers for their constructive criticism.
We hope that the manuscript is now suitable for publication.
We look forward to hearing from you,
Sincerely,
Corresponding author, on behalf of all authors.
The authors of this study evaluated the SARS-CoV-2 infection based on the previous metabolic status and further found its association with mortality and post-acute COVID-19.
The study included a population-based observational retrospective study conducted on a cohort of 110,726 patients aged 12 years or more, who were diagnosed with COVID-19 infection between June 1st 2021 and February 28th 2022 in the island of Gran Canaria. The authors reported in 347 patients who died, the combination of several confounding and risk factors was strongly predictive of mortality (p < 0.05).
The important part of the result obtained in 555 patients who developed post-acute COVID-19, the persistence of symptoms was most frequent in women, older subjects, and patients with obstructive sleep apnea, asthma or elevated basal glycemia (p < 0.05). The study concluded that elevated basal glycemia, low HDL-C, and high total cholesterol should be added to the list of well-known risk factors for severe COVID-19.
For the improvement of the manuscript, I am suggesting a few comments.
COMMENTS FOR REVIEWERS. MODIFICATIONS.
We thank the reviewers for their constructive comments that certainly improve the manuscript. The changes we have made to the manuscript are highlighted in yellow.
Point 1: Line 34 please include more references for the line
Response 1: We thank the reviewer for this suggestion. We have made these changes to the text. Please see the Introduction section (page 1, lines 36).
- Mantovani, A.; Byrne, C.D.; Zheng, M.H.; Targher, G. Diabetes as a risk factor for greater COVID-19 severity and in-hospital death: A meta-analysis of observational studies. Nutr. Metab. Cardiovasc. Dis.2020, 30, 1236–1248.
This study concluded that pre-existing diabetes is significantly associated with greater risk of severe/critical illness and in-hospital mortality in patients admitted to hospital with COVID-19.
- Korakas, E.; Ikonomidis, I.; Kousathana, F.; Balampanis, K.; Kountouri, A.; Raptis, A.; Palaiodimou, L.; Kokkinos, A.; Lambadiari, V. Obesity and COVID-19: Immune and metabolic derangement as a possible link to adverse clinical outcomes. Am. J. Physiol. Endocrinol. Metab.2020.
This study concluded that obesity has emerged as a major risk factor for worse COVID-19 outcomes. Chronic inflammation and oxidative stress, hypercytokinemia, immune dysregulation, endothelial dysfunction, and cardiovascular abnormalities are all possible mechanisms through which the excess in adipose tissue could lead to the acute hyperinflammatory state that characterizes severe SARS-CoV-2 infections and is responsible for its complications.
Point 2: Line 36-37 the author's study was also relating these variables so what is the novelty of this study here already they reported.
Response 2: We thank the reviewer for this suggestion.
This study showed how elevated baseline glycemia was risk factors for the development of post-acute COVID-19 (p < 0.05). High basal glycemia, low HDL-C and elevated total cholesterol were associated with a poorer COVID-19 prognosis and should be considered high-risk markers.
Other studies have already evaluated the association between alterations in analytical determinations and the patient's previous comorbidities and their association with a worse COVID outcome (admission to the ICU, mechanical ventilation, or death). However, until now the association between lipid profile alterations and basal glycemia in the development of post-acute covid had not been evaluated.
Please see the Discussion section (page 8, lines 228-235).
Point 3: Line 59 SARS COV- 2 better presents the full form.
Response 3: We thank the reviewer for this contribution. We have made these changes to the text. Further signs or symptoms like odynophagia, anosmia, ageusia, muscle pain, diarrhea, chest pain, chills, fatigue, nausea, and vomiting were also considered as symptoms of suspected SARS-CoV-2 infection, depending on the doctor’s criterion.
Please see the Materials and Methods section (page 2, lines 83-84).
Point 4: Line 80 In material and method the authors should write about the diagnostic approach to defining OSAS
Response 4: We thank the reviewer for this suggestion. We have made these changes to the text.
The participants were defined as known OSAS when there was a previous sleep study and/or the initiation of treatment documented by a physician.
Please see the Materials and Methods section (page 2, lines 70-72).
Point 5: In the material and method section, Patient consent has been taken or not, it should be written in the manuscript
Response 5: : We thank the reviewer for this suggestion.
Informed Consent Statement: Patient consent was waived due to anonymiza-tion/dissociation of patient data and the results did not affect the clinical management of patients.
Please see the Materials and Methods section (page 3, lines 126-128)
Point 6: Line 97-101 need to mention the appropriate reference
Response 6: We thank the reviewer for this contribution. We have made these changes to the text. Please see the Materials and Methods (Definitions) section (page 2, lines 76)
Touloumi, G.; on behalf of the EMENO study group; Karakosta, A.; Kalpourtzi, N.; Gavana, M.; Vantarakis, A.; Kantzanou, M.; Hajichristodoulou, C.; Chlouverakis, G.; Tryp-sianis, G.; et al. High prevalence of cardiovascular risk factors in adults living in Greece: The EMENO National Health Examination Survey. BMC Public Health 2020, 20, 1–10.
Point 7: Table 3 multivariate analysis please specify the factors in legends
Response 7: We thank the reviewer for this contribution. We have accepted the recommendation proposed by reviewer 2 to present the results of the logistic regression analysis in the forest plot.
We have specified the factors in the legend. Please see the Results section (pages 7-8).
Point 8: First para of discussion The best way to start any discussion is to present their own novelty research rather than discussing others, please modify
Response 8: We thank the reviewer for this contribution.
This study showed how high basal glycemia, low HDL-C and elevated total cholesterol were associated with a poorer COVID-19 prognosis and should be considered high-risk markers. The elevated basal glycemia was risk factors for the development of post-acute COVID-19.
Other studies have already evaluated the association between alterations in analytical determinations and the patient's previous comorbidities and their association with a worse COVID outcome (admission to the ICU, mechanical ventilation, or death) [6]. However, until now the association between lipid profile alterations and basal glycemia in the development of post-acute covid had not been evaluated.
Please see the Discussion section (page 8, lines 228-235).
Point 9: Line 197-198 is there any reason for this
Response 9: We thank the reviewer for this contribution.
The association between hypertension and worse out- comes of COVID-19 infection may be due to the higher frequency of comorbidities and a more advanced age of these individuals. An Italian cross-sectional study did not find hypertension as an independent factor affecting the outcome of COVID-19
Iaccarino, G.; Grassi, G.; Borghi, C.; Ferri, C.; Salvetti, M.; Volpe, M. Age and Multimorbidity Predict Death Among COVID-19 Patients: Results of the SARS-RAS Study of the Italian Society of Hypertension. Hypertension 2020, 76, 366–372.
Please see the Discussion section (page 9, lines 251-255).
Point 10: Line 236 Please cite those several meta-analyses
Response 10: We thank the reviewer for this contribution. We have made these changes to the text.
Similar findings were reported by Chen, et al. where female patients and patients with pre-existing asthma were more likely to develop a post-COVID-19 condition, with estimated pooled probability ratios (OR) of 1.57 (95%CI 1.09, 2.26) and 2.15 (95%CI 1.14, 4.05), respectively [24].
Please see the Discussion section (page 10, lines 293-296).
Point 11: Line 264-265 It is possible that the glycemic and lipidemic parameters are elevated in obese or overweight subjects, The authors need to check with different population's results and discuss.
Response 11: We thank the reviewer for this contribution.
It is possible that the glycemic and lipidemic parameters are elevated in obese or overweight subjects. There is a phenotype corresponding to individuals with normal weight but metabolically obese, that is, they have a normal BMI but present the typical alterations of obese patients: insulin resistance, low levels of HDL-C and high concentrations of triglycerides. At the same time, there are those who have been called metabolically healthy obese. These individuals have a BMI > 30, but none of the metabolic abnormalities typical of obese individuals.
Please see the Discussion section (page 10, lines 323-329).
Point 12: Line 284 What do the authors want to relate with COVID here
Response 12: We thank the reviewer for this contribution.
As shown in the present study, plasma lipid concentrations and basal glycemia should assist with the clinical management of COVID-19.
This difference in disease course can help to assess patients' prognosis and the occurrence of severe forms and thus provide optimal management.
Please see the Discussion section (page 11, lines 347-350).
Point 13: All references need to be modified based on the MDPI journals.
Response 13: We thank the reviewer for this contribution. We have modified all references based on the MDPI journals. Please see the References section.
Please see the attachment

Round 2
Reviewer 1 Report
Thank you. No further comments.
Best regards!
Author Response
We thank the reviewers for their constructive comments that certainly improved the manuscript.

Reviewer 2 Report
The authors have partially heeded my suggestions. I still think that a poisson regression model (the authors have not answered me correctly on this) would be more realistic with the ORs obtained. There is no doubt that they are overestimated.
PS: "0.000" should be replaced by "<0.001", not "0.001".
Kind regards,
Author Response
Response to Reviewer 2 Comments
Dear Mr. Reviewer,
Regarding the manuscript: "Assessment of SARS-CoV-2 infection according to previous metabolic status and its association with mortality and post-acute COVID-19" (Manuscript ID: nutrients-1815494)
Thank you very much for your feedback and for the opportunity to resubmit a second revised version for publication. We have addressed all issues and thank the reviewers for their constructive criticism.
We hope that the manuscript is now suitable for publication.
We look forward to hearing from you,
Sincerely,
Corresponding author, on behalf of all authors.
COMMENTS FOR REVIEWERS. MODIFICATIONS.
We thank the reviewers for their constructive comments that certainly improve the manuscript. The changes we have made to the manuscript are highlighted in yellow.
Point 1: PS: "0.000" should be replaced by "<0.001", not "0.001".
Response 1: We thank the reviewer for this contribution. We have replaced "p=0.001" with "p<0.001". Please see the Results section (tables).
Point 2: The authors have partially heeded my suggestions. I still think that a poisson regression model (the authors have not answered me correctly on this) would be more realistic with the ORs obtained. There is no doubt that they are overestimated.
Response 2: We thank the reviewer for this contribution.
We have decided to use a multivariable Poisson regression model adjusting for predefined covariates to estimate the propensity scores for cohort participants.
As a new find, in this model we found that the elevated total cholesterol was risk factors for the development of post-acute COVID-19.
In adittion, as the reviewer previously commented, the models used in the other version of the manuscript were overestimated.
Please see the Results section (figure 1 and 2).Please see the attachment.

Reviewer 3 Report
P=0.000 should be changed to <0.001.
"Participants were considered to have diabetes if they had glycemia ≥ 126 mg/dL or antidiabetic treatment." Do you mean fasting glucose levels?
When was information about subjects' metabolic status obtained?
Author Response
Response to Reviewer 3 Comments
Dear Mr. Reviewer,
Regarding the manuscript: "Assessment of SARS-CoV-2 infection according to previous metabolic status and its association with mortality and post-acute COVID-19" (Manuscript ID: nutrients-1815494)
Thank you very much for your feedback and for the opportunity to resubmit a second revised version for publication. We have addressed all issues and thank the reviewers for their constructive criticism.
We hope that the manuscript is now suitable for publication.
We look forward to hearing from you,
Sincerely,
Corresponding author, on behalf of all authors.
COMMENTS FOR REVIEWERS. MODIFICATIONS.
We thank the reviewers for their constructive comments that certainly improve the manuscript. The changes we have made to the manuscript are highlighted in yellow.
Point 1: P=0.001 should be changed to <0.001.
Response 1: We thank the reviewer for this contribution. We have replaced "p=0.001" with "p<0.001". Please see the Results section (tables).
Point 2: "Participants were considered to have diabetes if they had glycemia ≥ 126 mg/dL or antidiabetic treatment." Do you mean fasting glucose levels?
Response 2: We thank the reviewer for this contribution. In fact, we mean “fasting glucose levels”.
Please see the Materials and Methods section (page 2, lines 72-73).
Point 3: When was information about subjects' metabolic status obtained?
Response 3: We thank the reviewer for this contribution.
The information about subjects' metabolic status was obtained from the measurements made during the dates selected for the study period, that is, between June 1st 2021 and February 28th 2022. After the end of the study period, the last available measurement of the analytical determinations (total cholesterol, triglycerides, HDL-C, LDL-C and fasting glucose levels) and the anthropometric parameters (abdominal perimeter, body weight, height and BMI) was selected.
Please see the Materials and Methods section (page 3, lines 122-127).
Please see the attachment